# Inhibiting of self-renewal, migration and invasion of ovarian cancer stem cells by blocking TGF-β pathway

Haiyan Wen[1]*, Min Qian[1], Jing He[1], Meihui Li[1], Qing Yu[1], Zhengwei Leng[2]*

**1** Obstetrics Department, Hangzhou Women's Hospital, Hangzhou, Zhejiang, China, **2** General Surgical Teaching and Research Office, North Sichuan Medical College, Nanchong, Sichuan, China

* wenhy1991@163.com (HW); lengzhengwei@163.com (ZL)

## Abstract

### Objective

To investigate the effect and mechanism of SB525334 on self-renewal, migration and invasion of ovarian cancer stem cells.

### Methods

ALDHhigh-expressing cancer stem cells (CSCs) were isolated from human ovarian cancer cell line SKOV-3 by flow cytometry and treated with 2μg/mL SB525334 for 6h. The sphere forming assay was used to detect the ability of self-renewal of CSCs and the colony formation assay was used to detect the tumorigenicity *in vitro*. Transwell migration and invasion assay were used to detect the migration and invasion ability of CSCs. To further explore the mechanism, real-time quantitative PCR and flow cytometry were used to detect the mRNA and protein expression of TGF-β, Smad2, Smad3, phosphorylated Smad2, phosphorylated Smad3 and Smad4, respectively. Expressions of epithelial-mesenchymal transition (EMT)-related genes E-cadherin, Snail, Vimentin were also assessed.

### Results

The self-renewal ability, tumorigenicity *in vitro*, migration and invasion ability of CSCs were significantly attenuated after SB525334 treatment. The expressions of TGF-β, phosphorylated Smad2, phosphorylated Smad3, Snail, and Vimentin were decreased, while Smad4 and E-cadherin expressions were increased.

### Conclusion

SB525334 may inhibit the self-renewal, invasion and migration of ovarian CSCs by blocking the TGF-β/Smad/EMT pathway.

**Data Availability Statement:** All relevant data are within the paper.

**Funding:** The authors received no specific funding for this work.

Ovarian cancer is a malignant tumor that seriously threatens women's health, and the metastasis and recurrence of the tumor is the leading cause of death [1]. Numerous studies have

**Competing interests:** The authors have declared that no competing interests exist.

confirmed that a small group of cells, called cancer stem cells (CSCs) [2], with strong self-renewal ability and multi-directional differentiation ability in ovarian cancer mediates the tumor metastasis and recurrence. The molecular mechanism of CSCs self-renewal, invasion and metastasis has become a research hotspot and difficulty in cancer research [3, 4]. The investigation of the mechanism of ovarian cancer cell migration and invasion signaling pathway from the perspective of CSCs is also an important direction to reveal the ovarian cancer and metastasis. At present, researchers have classified ovarian cancer CSCs by labeling CD133, CD44, Aldehyde dehydrogenase (ALDH), Lgr5, SP, etc., while among them ALDH high expression (ALDHhigh) is considered to be the reliable surface marker [5–7].

The tumor metastasis recurrence is closely related to epithelial-mesenchymal transition (EMT) [8–10]. The tumor cells acquire more powerful drug-resistant, tumor-forming, and migration-invasive ability through EMT process. During this process, the expression of epithelial cell genes such as E-cadherin decreases and the expression of mesenchymal cell genes such as Snail, Slug, Vimentin increase [11, 12]. Studies have shown that the abnormal activation of TGF-β signaling pathway mediates the tumor metastasis and recurrence. The activation of TGF-β pathway has been found in breast cancer, non-small cell lung cancer and other tumors to promote EMT progression [10, 13, 14]. However, in ovarian cancer CSCs, whether TGF-β mediates the cell invasion and metastasis through EMT has not been reported. Therefore, in this study, ALDHhigh-expressing ovarian cancer CSCs will be selected by flow cytometry, and then TGF-β signaling pathway inhibitor SB525334 will be used on CSCs to detect the changes in malignant ability of cells together with the changes in TGF-β pathway and EMT key gene expression. The study is to reveal the role and mechanism of TGF-β in EMT-mediated invasion and metastasis of ovarian cancer CSCs.

# 1 Materials and methods

## 1.1 Materials

Human ovarian cancer cell line SKOV-3 (HTB-77) was obtained directly from ATCC and authenticated by short tandem repeat PCR profiling at ATCC. The cell lines were obtained in March 2016 and immediately cultured for experiments. The medium used to culture each cell line was according to the ATCC, and cells were cultured in a humidified incubator with 5% $CO_2$ at 37˚C. McCoy's 5a medium, DMEM/F12 medium, and fetal bovine serum were from Hyclone, USA. 7-amino-actinomycin D (7-AAD) was from Gibco, USA. Insulin, epidermal growth factor (EGF), reconstituted fibroblast growth factor (FGF), BSA, and B27 were from PeproTech, USA. TGF-β, phosphorylated Smad2, phosphorylated Smad3, Smad4, Snail, and Vimentin E-cadherin antibodies were from CST USA. ALDHhigh cell sorting kit was from Stem Cell, Canada. TGF-β pathway inhibitor SB525334 was from R&D, USA. All primers were synthesized by Invitrogen, USA.

## 1.2 Methods

**1.2.1 Cell culture.** SKOV-3 cells were cultured in a cell culture incubator at 37 ˚C with 5% $CO_2$ using McCoy's 5a medium containing 10% fetal bovine serum, and the logarithmic growth phase cells were chosen to subsequent experiments. The sorted ALDHhigh CSCs were cultured in serum-free DMEM/F12 medium containing 5 mg/mL insulin, 0.4% BSA, 2% B27, 10 ng/mL FGF, and 10 ng/mL EGF for 48 h and then subjected to subsequent experiments.

**1.2.2 ALDHhigh CSCs sorted by flow cytometry.** Cell sorting was performed according to the instructions of Aldefluor assay kit. SKOV-3 single cells were obtained by trypsin digestion, incubated with ALDEFLUOR reagent for 50 min at 37 ˚C, and the control cells were incubated by ALDEFLUOR inhibitor DEAB to determine the flow cytology of the ALDHhigh

CSCs. 2 μg/mL 7-AAD was added to stain at room temperature for 10 min to remove the dead cells during sorting. Then BD FACSAria Fusion flow cytometry was used to obtain CSCs with ALDHhigh expression [15].

**1.2.3 Sphere forming assay.** 500 cancer stem cells were inoculated in serum-free DMEM/F12 medium containing 5 mg/mL insulin, 0.4% BSA, 2% B27, 10 ng/mL FGF and 10 ng/mL EGF for 96 h. The number of spheroids was counted under a microscope and statistical analysis was performed.

**1.2.4 Plate colony formation assay.** 200 cancer stem cells were inoculated into 6-well plates of McCoy's 5a medium containing 10% fetal bovine serum for 7 days, rinsed twice with PBS, fixed in 75% ethanol for 15 min at room temperature, stained with crystal violet at room temperature for 15 min, then rinsed 4 times with PBS. Finally, they were photographed and counted for colony formation for statistical analysis.

**1.2.5 Transwell migration and invasion experiment.** $4\times10^4$ cells were suspended in 50 μL of serum-free RPMI-1640 medium and placed in the upper chamber of Transwell chamber with 8 μm pore size filter. The chamber contained 100 μL RPMI-1640 with 10% FBS and was laid with Matrigel (for invasion experiment) or without Matrigel (for migration experiment), and the cells were incubated for 13 h in the cell culture incubator. The cells that migrated through the filter were stained with 0.1% crystal violet for 15 min at room temperature, and photographed under the microscope for statistics.

**1.2.6 qRT-PCR experiment.** The detail experimental method can be found in our previous works [15]. Trizol method was used to extract RNA from cells. After reverse transcription and PCR amplification, the relative expression of TGF-β signaling pathway and EMT key gene mRNA would be detected, while GAPDH is used as an internal reference and $2^{-\Delta\Delta Ct}$ is the relative expression level of the gene. The primer sequences are as follows:

GAPDH: 5'-GGAGCGAGATCCCTCCAAAAT-3',

5'-GGCTGTTGTCATACTTCTCATGG-3';

TGF-β: 5'-TGGAGCAACATGTGGAACTC-3',

5'-CAGCAGCCGGTTACCAAG-3';

Smad2: 5'-GATGCTCTGGCGTCTACT-3',

5'-GACACCCAATTCCTTCAC-3';

Smad3: 5'-AAACCAGGCTGGGTAAACAAGTG-3',

5'-GCAACAGCAGCAGTGAAGGTG-3';

Smad4: 5'-ATGACCTTCGTCGCTTATGC-3',

5'-GGCCCGGTGTAAGTGA-3';

E-cadherin: 5'- GCCCTGCCAATCCCGATGAAA-3',

5'- GGGGTCAGTATCAGCCGCT-3';

Snail: 5'- CCAGACCCACTCAGATGTCAAGAA-3',

5'- GGCAGAGGACACAGAACCAGAAAA-3';

Vimentin: 5'- GCTTCAGAGAGAGGAAGCCGAAAA-3',

5'-CCGTGAGGTCAGGCTTGGAAA-3'.

**1.2.7 Flow cytometry to detect TGF-β pathway and EMT gene protein expression.** $3\times10^5$ cells were fixed at room temperature for 40 min using BD transdermal solution, and then incubated with 1/200 ratio of primary antibody for 40 min at 4 ˚C in the dark. After washed with cold PBS, they were incubated with PE-labeled 1:200 secondary antibody for 30 min at 37 ˚C. For comparison, the isotype control samples were incubated directly with 1:200 secondary antibody without adding primary antibody. All the data were analyzed using FlowJo 10.0 software for analysis. The average fluorescence intensity ratio is proportional to the amount of protein expression, which can be obtained from the ratio of mean fluorescence intensity of the samples to the corresponding average fluorescence intensity of the isotype control samples.

## 1.3 Statistical analysis

Statistical analysis was performed using SPSS 19.0 software. The normal distribution measurement data were expressed by $\bar{x} \pm s$, and the comparison between groups was performed by paired sample t test. $P<0.05$ was considered statistically significant.

# 2 Results

## 2.1 Identification of ALDHhigh cells

We first sorted out ALDHhigh and ALDHlow cells (Fig 1A), and then used sphere forming assay and plate colony formation assay to identify tumor stem cell characteristics. The results show that ALDHhigh and ALDHlow cells have the spheroid numbers of 345±19.52 and 57.67 ±7.09 (t = 23.96, P = 0.0001) (Fig 1B); the numbers of colony formation are 46±4.58 and 26.67 ±2.08 (t = 6.65, P = 0.0027), respectively (Fig 1C). Thus it is suggested that ALDHhigh is a reliable cell model for studying ovarian cancer CSCs.

## 2.2 Effect of SB525334 on the self-renewal ability of CSCs

According to the previous research results and literature reports [9], we used 2 μg/mL SB525334 to act on CSCs for 6 h and then carried out the follow-up study (experimental group). The sphere forming assay result shows that the cell spheroid numbers in the experimental group and the control group are 126±16 and 355±9 (t = 21.61, P = 0.0001), suggesting that SB525334 can significantly impair the self-renewal ability of CSCs (Fig 2A).

## 2.3 Effect of SB525334 on the colony formation ability of CSCs

The plate colony formation assay shows that the numbers of clones in the experimental group and the control group are 15±4.58 and 52.33±6.81 (t = 7.88, P = 0.0014), respectively, suggesting that SB525334 can significantly impair in vitro colony formation ability of CSCs (Fig 2A).

## 2.4 Effect of SB525334 on the migration and invasion ability of CSCs

We used Transwell migration and invasion assay to detect the migration and invasion ability of the two groups. The numbers of migrated cells in the experimental group and the control group are 19.67±6.03 and 74.33±4.16 (t = 12.93, P = 0.0002) (Fig 2B); while the corresponding numbers of invasive cells are 8.67±2.08 and 69.33±8.50 (t = 12.0007, P = 0.0003) (Fig 2C). The data show that SB525334 can significantly inhibit the cell migration and invasion of ovarian cancer CSCs.

## 2.5 Effect of SB525334 on TGF-β pathway in CSCs

The mechanism of SB525334 inhibition of CSCs self-renewal, clonality, and migration invasiveness is also investigated. mRNA and protein expression changes of the key genes of TGF-β

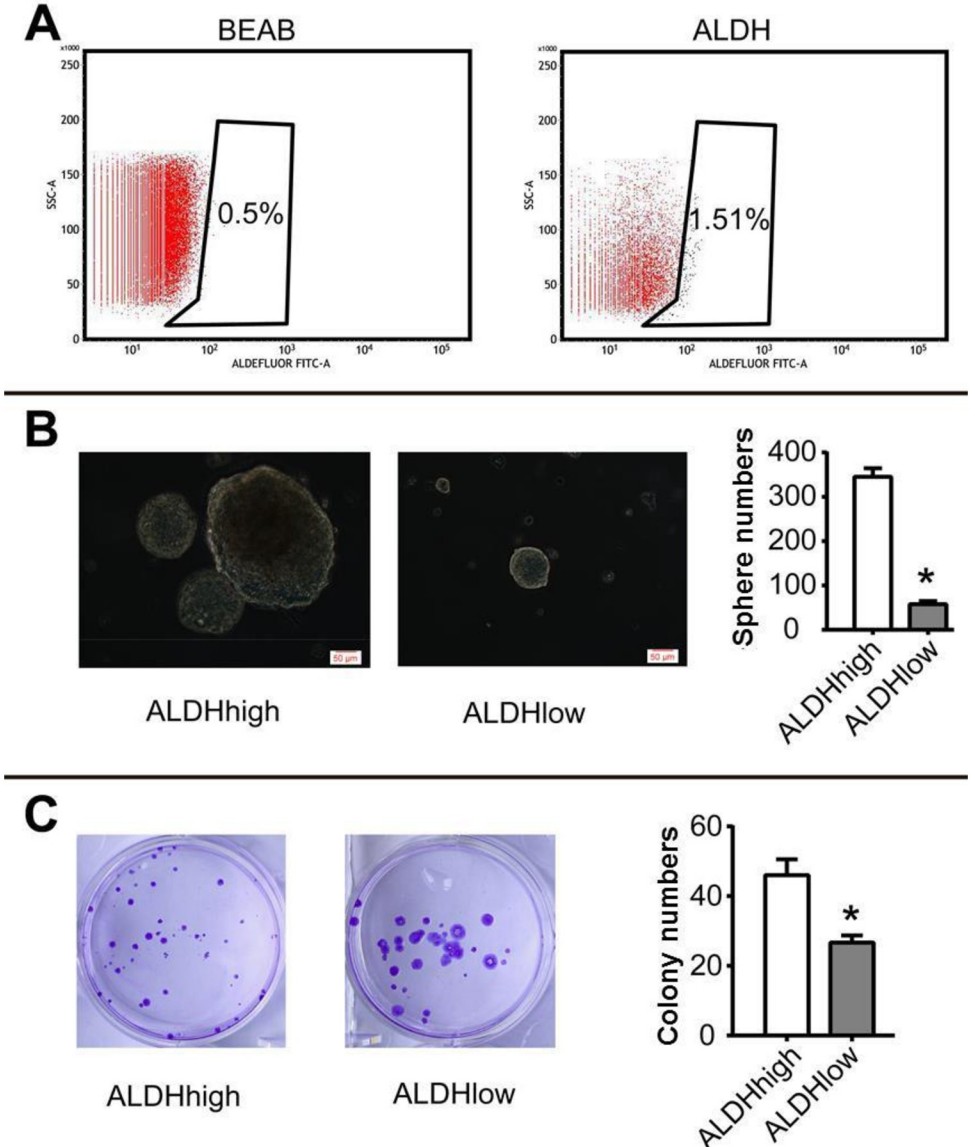

**Fig 1. Identification of ALDHhigh tumor cells.** (A) Flow cytometry sorting of ALDHhigh and ALDHlow tumor cells; (B) Sphere forming ability of ALDHhigh and ALDHlow cells; (C) Plate colony formation ability of ALDHhigh and ALDHlow cells; Scale bar: 50 μm; *P <0.05.

pathway as TGF-β, Smad2, Smad3, phosphorylated-Smad2 (p-Smad2), phosphorylated-Smad3 (p-Smad3), and Smad4 were detected by qRT-PCR and flow cytometry, respectively. qRT-PCR results show (Fig 3A) that the relative mRNA expression levels of TGF-β, Smad2, Smad3, and Smad4 are 0.46±0.06, 1.07±0.15, 1.13±0.21, and 3.7±0.53 in the experimental group, while they are 1.03±0.06, 1.02±0.1, 1.04±0.07, and 1.05±0.14 in the control group, respectively (t = 11.873, 0.480, 0.741, 8.441; P = 0.0003, 0.6564, 0.5, 0.0011). The protein data of flow cytometry show (Fig 3A and 3B) that the relative expression levels of TGF-β, Smad2, p-Smad2, Smad3, p-Smad3 and Smad4 in the experimental group and the control group are 3.54 ±0.25, 16.95±1.68, 6.74±1.05, 14.27±0.9, 6.46±0.5, 10.4±0.84 and 13.88±0.29, 17.61±0.76, 14.62 ±1.13, 16.74±1.03, 3.5±0.36 (t = 46.57, 0.61, 8.82, 2.47, 15.53, 13.11; P = 0.0001, 0.5724, 0.0009,

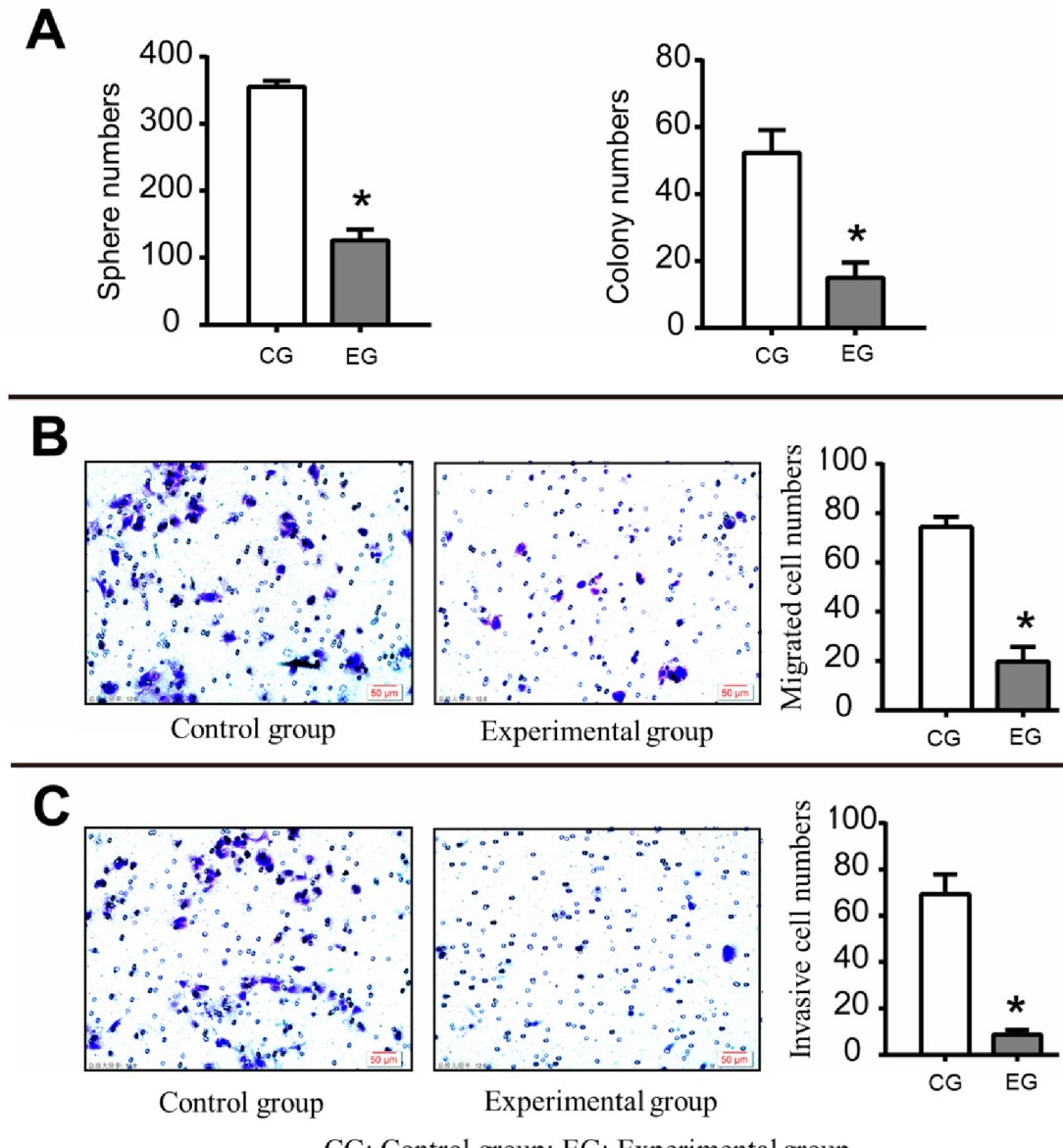

**Fig 2. Effect of SB525334 on the malignant behavior of CSCs.** (A) The spheres numbers and colony numbers for experimental group and control group (The cells' self-renewal ability and colony forming ability in the experimental group are significantly weakened); (B) The migration ability of CSCs (The migration ability of CSCs in the experimental group is significantly weakened); (C) The invasion ability of CSCs (It is weakened in the experimental group); Scale bar: 50 μm; *P <0.05.

0.0688, 0.0001, 0.0002), respectively. The results confirm that SB525334 may inhibit the self-renewal, colony formation and migration invasiveness of ovarian cancer CSCs by inhibiting the activity of TGF-β pathway.

## 2.6 Effect of SB525334 on EMT gene expression in CSCs

Studies have shown that activation of TGF-β pathway will promote the EMT progress, leading to invasion and migration of tumor cells. However, such phenomenon has not been reported in ovarian cancer CSCs. Therefore, we further studied the effect of SB525334 on EMT gene

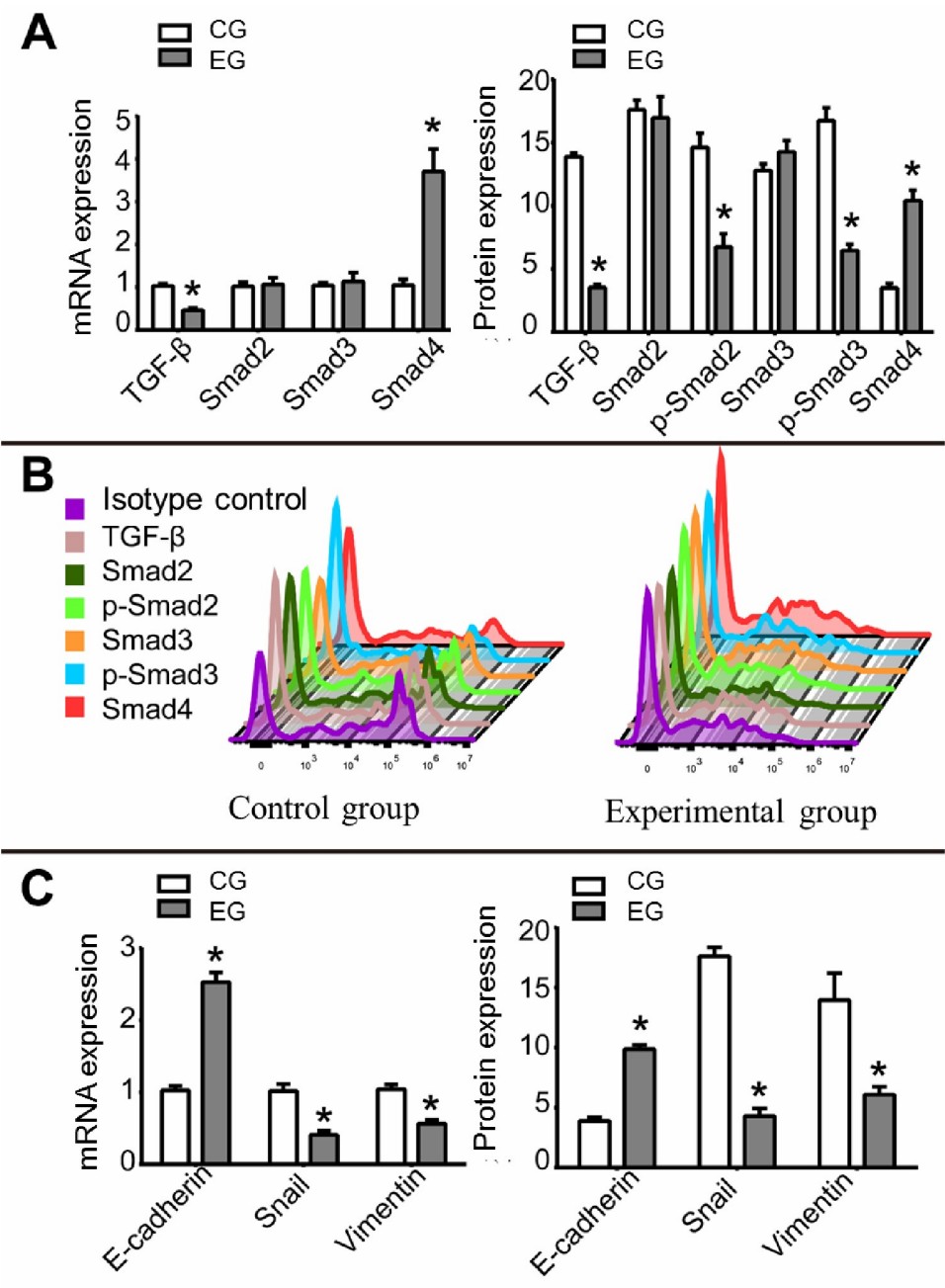

**Fig 3. Effect of SB525334 on TGF-β pathway and EMT in CSCs.** (A) mRNA expression (A) and protein expression (B) levels of TGF-β, Smad2, p-Smad2, Smad3, p-Smad3, and Smad4 in CSCs with and without the use of SB525334; (C) mRNA and protein expressions of E-cadherin, Snail and Vimentin in CSCs with and without the use of SB525334; *P <0.05.

expression in ovarian cancer CSCs. The qRT-PCR results (Fig 3C) show that the relative mRNA expression levels of E-cadherin, Snail and Vimentin in the experimental group and the control group are 2.53±0.14, 0.41±0.06, 0.56±0.06 and 1.03±0.06, 1.02±0.1, 1.04±0.07 (t = 17.45, 9.54, 9.64; P = 0.0001, 0.0007, 0.0006), respectively. The protein data from flow

cytometry (Fig 3C) show that the relative protein expression levels of E-cadherin, Snail and Vimentin in experimental group and the control group are 9.87±0.35, 4.29±0.65, 6.07±0.67 and 3.88±0.29, 17.61±0.76, 13.95±2.25 (t = 22.81, 23.12, 5.81; P = 0.0001, 0.0001, 0.0044), respectively. The results suggest that the EMT process of ovarian cancer CSCs is inhibited.

## 3 Discussion

In this study, we have confirmed that the self-renewal and clonality of cells are significantly inhibited by the application of SB525334 in CSCs through sphere forming and plate colony formation assays. The ability of cells to migrate and invade is one of the most important factors determining the migration and invasion of CSCs. We have found that after the treatment of TGF-β pathway inhibitor SB525334, the migration and invasion ability of CSCs decreases significantly, suggesting that the inhibitor SB525334 may effectively inhibit the TGF-β pathway and lead to the weakening of CSCs migration and invasion ability.

To further reveal the molecular mechanism of TGF-β affecting cell malignancy in ovarian cancer CSCs, the key gene expression of TGF-β pathway and its downstream EMT are also measured by qRT-PCR and flow cytometry. The results show that the expressions of TGF-β, p-Smad2 and p-Smad3 decrease and Smad4 increases after the treatment of SB525334 on CSCs. And the expressions of Snail and Vimentin in interstitial cells decrease and the expression of E-cadherin in epithelial cells increases. These outcomes suggest that SB525334 can inhibit the phosphorylation of Smad2/3 and lead to a decrease in the activity of TGF-β pathway, further inhibiting the EMT process of CSCs, which results in the impairment of the self-renewal, colony formation and migration of CSCs. This study may provide clinical diagnostic indicators or potential therapeutic targets for ovarian cancer CSCs-mediated tumor metastasis and recurrence.

## Author Contributions

**Conceptualization:** Haiyan Wen, Zhengwei Leng.

**Data curation:** Jing He, Meihui Li, Qing Yu.

**Formal analysis:** Jing He.

**Investigation:** Haiyan Wen, Min Qian.

**Methodology:** Min Qian.

**Project administration:** Haiyan Wen.

**Writing – original draft:** Haiyan Wen, Meihui Li, Zhengwei Leng.

**Writing – review & editing:** Haiyan Wen.

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
