## [Decision Letter · Decision Letter 0]

26 Feb 2020

Inhibiting of self-renewal, migration and invasion of ovarian cancer stem cells by blocking TGF-β pathway

PONE-D-19-33395

Dear Dr. Wen,

We are pleased to inform you that your manuscript has been judged scientifically suitable for publication and will be formally accepted for publication once it complies with all outstanding technical requirements.

With kind regards,

Salvatore V Pizzo

Academic Editor

PLOS ONE

1. Please provide additional information about each of the cell lines used in this work, including source, history, culture conditions and any quality control testing procedures (authentication, characterisation, and mycoplasma testing). For more information, please see http://journals.plos.org/plosone/s/submission-guidelines#loc-cell-lines

Reviewers' comments:

Reviewer's Responses to Questions

**Comments to the Author**

1. Is the manuscript technically sound, and do the data support the conclusions?

Reviewer #1: Yes

2. Has the statistical analysis been performed appropriately and rigorously? 

Reviewer #1: I Don't Know

3. Have the authors made all data underlying the findings in their manuscript fully available?

Reviewer #1: Yes

4. Is the manuscript presented in an intelligible fashion and written in standard English?

Reviewer #1: Yes

5. Review Comments to the Author

Reviewer #1: I reviewed the manuscript ''Inhibiting of self-renewal, migration and invasion of ovarian cancer stem cells by blocking TGF-β pathway'' by Wen et al. I appropriate the enormous effort made by authors in performing and writing this original article. This is a well-written and well organized paper with fluent language and rational structures. Therefore, I recommend its publication in the PLOS ONE.

6. PLOS authors have the option to publish the peer review history of their article (what does this mean?). If published, this will include your full peer review and any attached files.

Reviewer #1: No

---

## [Editor Report · Acceptance letter]

12 Mar 2020

PONE-D-19-33395 

Inhibiting of self-renewal, migration and invasion of ovarian cancer stem cells by blocking TGF-β pathway 

Dear Dr. Wen:

I am pleased to inform you that your manuscript has been deemed suitable for publication in PLOS ONE. Congratulations! Your manuscript is now with our production department. 

With kind regards,

on behalf of

Dr. Salvatore V Pizzo 

Academic Editor

PLOS ONE